# Fermented Wheat Germ Protein with Histone Deacetylase Inhibitor AR42 Demonstrates Enhanced Cytotoxicity against Lymphoma Cells In Vitro and In Vivo

**DOI:** 10.3390/ijms25147866

**Published:** 2024-07-18

**Authors:** Joshua F. Meckler, Daniel J. Levis, Yanguo Kong, Robert T. O’Donnell, Daniel P. Vang, Joseph M. Tuscano

**Affiliations:** 1Division of Hematology and Oncology, Department of Internal Medicine, University of California Davis School of Medicine, Sacramento, CA 95817, USA; jfmeckler@ucdavis.edu (J.F.M.); danjlevis@yahoo.com (D.J.L.);; 2Department of Veterans Affairs, Northern California Healthcare System, Sacramento, CA 95652, USA

**Keywords:** lymphoma, fermented wheat germ, histone deacetylase, AR42, REC2282

## Abstract

Current treatments for lymphoma are plagued by substantial toxicity and the inability to overcome drug resistance, leading to eventual relapse and rationalizing the development of novel, less toxic therapeutics and drug combinations. Histone deacetylase inhibitors (HDACis) are a broad class of epigenetic modulators that have been studied in multiple tumor types, including lymphoma. Currently, HDACis are FDA-approved for treating relapsed T-cell lymphomas and multiple myeloma, with ongoing trials in other lymphomas and solid tumors. As single agents, HDACis frequently elicit toxic side effects and have limited efficacy; therefore, many current treatment strategies focus on combinations to boost efficacy while attempting to minimize toxicity. Fermented wheat germ extract (FWGE) is a complementary agent that has shown efficacy in several malignancies, including lymphoma. Here, we utilize a more potent FWGE derivative, known as fermented wheat germ protein (FWGP), in combination with the HDACi AR42, to assess for enhanced activity. We report increased in vitro killing, cell cycle arrest, and in vivo efficacy for this combination compared to each agent alone with minimal toxicity, suggesting a potentially new, minimally toxic treatment modality for lymphoma.

## 1. Introduction

Despite significant advances in treatments for lymphoma, most of the patient population will ultimately relapse and succumb to the disease [1,2,3]. Additionally, concerns with treatment-related toxicities abound, especially for the elderly population who often discontinue treatment due to side effects. One natural product preparation that has shown activity against various malignant cell lines both in vitro and in vivo, without appreciable side effects, is FWGE [4,5,6,7,8,9]. FWGE is comprised of hundreds or more components, including several, such as benzoquinones and agglutinin, that are known to be cytotoxic to cancer cells [6,8]. Preliminary in vitro studies suggest that FWGE may downregulate major histocompatibility complex class I in lymphoma and modulate cell cycle arrest and apoptosis in pancreatic carcinoma, hepatocellular, and leukemia cell lines [4,5,9,10,11,12]. In addition, FWGE has been shown to have selective inhibitory and inducing effects on glycolysis and pentose phosphatase enzymes, respectively, leading to decreased nucleotide synthesis correlating with cell cycle arrest in a variety of cancer cell lines [6,10]. Pairing FWGE or its components with chemotoxic treatments has demonstrated enhanced therapeutic efficacy often at lower doses, suggesting synergy with various antineoplastics [8,9,13]. For example, treatment of isolated B cells from healthy donors in vitro with FWGE required over 2× the dose to reach IC_50_ when compared to lymphoma cell lines, suggesting cancer specificity [4]. Further, treatment of immunocompromised mice with daily doses of up to 1000 mg/kg of FWGE resulted in no decreased body weight over the course of 12 weeks, suggesting the low probability of toxicities across a broad therapeutic range. More recent work has elucidated a protein formulation of FWGE, known as fermented wheat germ protein (FWGP), that has more potent direct cytotoxicity against lymphoma via apoptosis and mediates enhanced natural killer (NK) cell-mediated cytotoxicity [4]. Moreover, recent studies have also demonstrated in vivo anti-tumor activity against lung adenocarcinoma [14]. Initial work with FWGP in lymphoma has expanded on the mechanistic understanding of FWGE, including effects on cell cycle arrest with rapid upregulation of cyclin-dependent kinase inhibitor CDKN1A, and induction of anti-proliferative markers p53, BAD, and BAK [4].

HDACis are a broad class of molecules that inhibit various histone deacetylase enzymes, inducing open chromatin conformations that confer changes in the expression of several genes related to the cell cycle [15,16,17,18]. Early FDA-approved HDACis include the monotherapies vorinostat and romidepsin for the treatment of T-cell lymphomas. Presently, HDACis are in clinical trials as combination therapies across solid and liquid tumors [18]. AR42 is a bicyclic benzamide class 1 and 2b histone deacetylase inhibitor that has demonstrated anti-tumor efficacy in various mammalian models in vivo and in vitro [16,17,19,20,21]. AR42 has been studied in phase 1 trials for the treatment of multiple myeloma, B-, and T-cell malignancies, and currently has FDA fast-track designation (under the commercial name REC2282), in a phase 2/3 trial for *NF2*-driven tumors [22,23,24]. While early efficacy data in lymphoma is modest, higher doses of AR42 may produce side effects, including high rates of grade 3–4 neutropenia resulting in treatment reductions that can affect efficacy. Despite similar treatment toxicities, another class 1 benzamide HDACi, entinostat, is currently in phase 3 trials as combination therapy with PD-1 and HER2 inhibitors in melanoma and breast cancer, respectively, suggesting that favorable strategies exist in combination regimens [25,26]. Due to their epigenetic modulating nature, HDACis such as AR42 are often used in combination with other therapies to achieve an enhanced response [20,21,27]. In multiple pre-clinical lymphoma models, AR42 combination treatment demonstrated enhanced efficacy compared to single agents, with lower IC_50_ values potentially translating to lower toxicities in the clinic [28,29].

Multiple molecular mechanisms with AR42 treatment of malignancies have been identified, and current evidence suggests conservation across tumor types. In both patient-derived and immortalized leukemia cell lines, AR42 demonstrated both class 1 and 2 HDACi activity through persistent acetylation of histone H3 (HDAC class 1) and tubulin (HDAC class 2b) as detected by Western blot analysis, while in multiple myeloma cell lines only H3 acetylation has been demonstrated [17,30]. These initial HDACi-mediated events trigger a broad range of downstream processes including cell cycle arrest, apoptosis, and DNA repair inhibition. In lymphoma and multiple myeloma cell lines treated with AR42, increases in cyclin-dependent kinase (CDK) inhibitor protein p21 and annexin V cell populations were demonstrated, suggesting cell cycle arrest and induction of apoptosis [16,28]. Treatment with AR42 resulted in dose-dependent cleavage of caspases 3, 8, and/or 9—suggesting an extrinsic apoptotic activation pathway—and downregulation of anti-apoptotic markers c-FLIP and BCL in various leukemia cell lines [16,17,30]. Further, treatment with AR42 resulted in increased PARP cleavage, indicating inhibition of a DNA repair mechanism, often upregulated in fast-dividing cancer cells. Additionally, in vitro data from treatment with AR42 supports several similar apoptotic and anti-proliferative mechanisms in melanoma, ovarian, bladder, and colon cancer [19,20,21,31], while treatment of normal ovarian surface epithelial (NOSE) cells with AR42 required a significantly higher dosage to reach in vitro IC_50_ [31]. Interestingly, in ovarian cancer cells, treatment with AR42 similarly demonstrated sustained acetylation of H3 and tubulin, suggesting the activity as a class 1/2b HDAC is conserved between solid and liquid tumors [31].

Here, we present an initial study examining the in vitro and in vivo activity of the combination of FWGP and AR42, demonstrating the efficacy of the combination without appreciable toxicity.

## 2. Results

### 2.1. Combination of AR42 with FWGP Lowers IC_50_ Values for AR42

Raji cells were incubated with either FWGP (20 µg/mL), AR42 (0.2 µM), AR42 + FWGP, or media only for 24, 48, or 72 h, and viability was measured using the MTS assay (Figure 1A). At each time point, the combination treatment yielded a significant reduction in tumor cell viability when compared to either treatment alone, with the greatest difference resulting after 24 h (51% growth inhibition with combination treatment vs. 10% or 15% with FWGP and AR42, respectively). All treatments yielded >60% growth inhibition after 72 h; however, the combination treatment inhibited growth at a rate of 82% vs. 65% and 75% for FWGP and AR42, respectively (Figure 1A). In vitro growth inhibition with FWGP and AR42 monotherapy was consistent with prior studies [4,28,29]. Additionally, the IC_50_ values for treatment with AR42 + FWGP were lower than each monotherapy in all but one cell line tested (DOHH2) (Figure 1B,C). The relative reductions in IC_50_ values compared to AR42 monotherapy per cell line were as follows: 43% (CA46), 60% (Daudi), 69% (DG75), 84% (Raji), 23% (Ramos), and 46% (SuDHL-4). All combination treatments that resulted in a reduction in IC_50_ value for AR42 were classified as additive using the combination index (CI) metric from isobologram analysis.

### 2.2. Cell Cycle Analysis in FWGP and AR42 Treated Raji Cells

We investigated potential perturbations to cell division with treatments of FWGP and/or AR42. After 24 h, treatment groups yielded higher proportions of cell populations in the G0/G1 phase when compared to untreated controls (Figure 2; Untreated = 38.81%, FWGP = 50.51%, AR42 = 51.11%, and FWGP + AR42 = 68.79%), suggesting a slowing of cellular division. Further, a higher proportion of cells presented in the G2/M phase in the untreated group when compared to treatment groups (Untreated = 47.69%, FWGP = 32.89%, AR42 = 32.99%, FWGP + AR42 = 27.46%). Interestingly, only the combination treatment showed a reduction in cell proportion in the S phase when compared to the untreated control (Untreated = 13.5%, FWGP + AR42 = 3.75%). Histogram plots used to calculate cell cycle proportions are presented in Figure 2.

Additionally, we investigated mechanisms of cellular killing with AR42 and FWGP using agents for the detection of early apoptotic, necrotic/late apoptotic, and dead Raji cells (Annexin V+, 7-AAD+, or both). Figure 3 demonstrates a time- and dose-dependent increase for the necrotic and dead Raji cell populations with AR42 and/or FWGP treatment. After 72 h of incubation, cells treated with the combination AR42 (0.2 µM) and FWGP (20 µg/mL) showed a significant increase in overall cytotoxicity (dead + necrotic) when compared to either agent alone (*p* = 0.003) with nearly a two-fold increase in killing. Importantly, killing was similar between both AR42 and FWGP monotherapy, suggesting equal contributions when combined. The mechanism of this increased killing involves a combination of apoptotic and necrotic induction. Finally, cells positive for Annexin V only (early apoptosis) showed no appreciable increase with combination therapy at either FWGP dose (20 or 5 µg/mL). Interestingly, while the early apoptotic population for AR42 alone treatment increased from 6.4 to 26.2 to 52.9% at 24, 48, and 72 h, respectively, many of these early apoptotic cells seemed to be converted to necrotic and/or dead cells (i.e., 7-AAD+) via the addition of FWGP. This was especially apparent at 72 h, where AR42 alone caused 52.9% of cells to enter early apoptosis, while the combination of AR42 and FWGP (20 µg/mL) resulted in only 16.1% of cells staining in the early apoptotic quadrant but 39% of cells staining dead + necrotic (Appendix A).

### 2.3. In Vitro FWGP Treatment Inhibits Deacetylase Activity and Induces Cell Cycle Inhibition

Prior work has demonstrated that FWGP treatment upregulates mRNA transcripts for cyclin-dependent kinase inhibitor molecules CDKN2A and CDKN1B [4]. In order to assay the protein levels of these same intrinsic cell cycle elements, FWGP-treated and control Raji cells were processed by MD Anderson’s Functional Proteomics Reverse Phase Protein Array Core facility. At low doses of FWGP, modest induction of CDK1B and CDKN2A was seen as early as 1 h post-treatment and increased with dose (Figure 4A).

Additionally, Western blot analysis detecting acetylated tubulin was performed to measure the inhibitor effect of both FWGP and AR42 on histone deacetylase. Figure 4B demonstrates the modest increase in acetylated tubulin with FWGP + AR42 combination with longer incubation (24 h). Raw Western blot images are presented in the Appendix A.

### 2.4. Treatment of Lymphoma Xenograft Mice with FWGP + AR42 Yields Additive Tumor Growth Inhibition

To assess the efficacy of tumor growth inhibition, Raji xenografts were established in an immunocompromised murine model prior to treatment initiation. Treatment doses of AR42 and FWGP were selected based on comparable doses from prior studies [4,29]. After 45 days of alternate daily treatment, the average tumor volume in the treatment groups was significantly less than the untreated control (Figure 5A; Untreated = 1996.75 ± 530 mm^3^, AR42 = 789.60 ± 278 mm^3^, FWGP = 876.58 ± 336 mm^3^, and FWGP + AR42 = 358.50 ± 126 mm^3^). Interestingly, the combination treatment of FWGP with AR42 resulted in >50% greater inhibition of average tumor growth compared to either treatment alone, suggesting a potential synergistic effect. No treatments yielded significant toxicity requiring treatment discontinuation (Appendix A). Additionally, tumor weight was analyzed at the conclusion of the study. Treatment groups yielded average tumor weight that was markedly reduced compared to the untreated group (Figure 5B; Untreated = 1546 ± 890 mg, FWGP = 930 ± 785 mg, AR42 = 465 ± 309 mg, FWGP + AR42 = 384 ± 205 mg). Photographs demonstrating tumor growth inhibition before and after tumor resection are presented in Appendix A.

## 3. Discussion

Here, we have investigated a novel lymphoma treatment combination of the HDAC inhibitor AR42 and FWGP. When compared to either agent alone, the combination of FWGP with AR42 reduced IC_50_ values in all but one lymphoma cell line tested in vitro. Current clinical studies of AR42 and another HDACi in the same class, entinostat, report high rates of grade 3/4 neutropenia and thrombocytopenia requiring dose reductions based upon treatment toxicities [22,23,32]. In these early clinical studies with AR42, patients receiving the highest dose experienced dose-limiting toxicities, thus indicating a need for more potent HDACis at lower doses. In contrast, we are unaware of any studies documenting significant treatment-related toxicity of FWGE or FWGP in vivo or in the limited clinical trials published [33,34,35]. Thus, combination regimens with FWGP that allow for a lower dose of AR42 while maintaining or exceeding monotherapy efficacy offer a potential benefit to patients without additional toxicity. Indeed, the in vivo dose used in this current study is dwarfed by those from most other pre-clinical studies investigating AR42, including a study using the same xenograft model [17]. Several studies have verified the efficacy of FWGE in vitro and in vivo, and our group has previously reported the efficacy of its protein formulation, FWGP, in lymphoma and lung cancer murine models in vivo [4,14]. While several components have been identified in FWGP, the components that mediate anti-cancer properties have not yet been fully elucidated. It has been hypothesized that benzoquinones, agglutinins, fiber, lipids, and wheat bran phytic acid may play an anti-cancer role. However, prior work has demonstrated that FWGP fractions maintain in vitro cytotoxicity, immune modulation, and in vivo tumor inhibition even in the absence of known cytotoxic elements wheat germ agglutin (WGA) or benzoquinones [4]. Additionally, treating FWGE with heat or proteinase K to remove or denature proteins resulted in decreased in vitro cytotoxicity, thus further suggesting a protein or peptide component being responsible for the anti-cancer activity.

Despite the present uncertainty about the exact active component(s) within FWGP, previous data have suggested several mechanisms to explain the in vitro cytotoxicity and in vivo efficacy of FWGP. For FWGP, these include induction of NK cell-mediated cytotoxicity (as evidenced by decreased efficacy after NK cell depletion), direct pro-apoptotic-mediated cytotoxicity via activation of multiple caspases, and modulation of BAK/BAD-dependent apoptotic pathways [4]. Previous studies that examined the treatment of various cancer cell lines with the FWGP parent product, FWGE, demonstrated induction of cell cycle arrest, induction of pentose phosphate pathway through a reduction in glycolysis enzymes, change in mitochondrial metabolism, and an increase in oxidative stress markers in cancer cells [5,6,9,10,11,36]. Our findings here further support tumor suppression through cell cycle arrest and are consistent with prior work demonstrating halted cell division in the G_0_/G_1_ phase after FWGE/FWGP treatment [5,6,10,14]. It is possible that modulation of such pathways is enhanced during combination treatment with AR42, which has demonstrated a reduction in cyclin D1 through p16 and p21 induction [16,20]. Coupled with prior mechanistic data, our findings herein suggest a potential conserved mechanistic pathway between AR42 and FWGP that includes cell cycle arrest through induction of CDKN2A (p21) or CDKN1B and HDACi activity measured through increased acetylated tubulin. Despite this initial hypothesis, studies investigating apoptosis and necrosis yielded a greater portion of AR42 + FWGP treated cells shifted to the dead/necrosis phase when compared to AR42 monotherapy—suggesting FWGP + AR42 may mediate quicker progression of cells from early apoptosis to death after 72 h compared to either agent alone. Given that prior studies have cataloged pro-apoptotic markers BAX/BAD and increased Annexin V cell populations with FWGP at doses of 200 µg/mL we predict the present results to reflect 10× dose reduction in FWGP. Taken together, the modulation of multiple anti-tumor mechanisms suggests a treatment benefit with the combination of FWGP and AR42. As is common with broad HDACis, AR42 may not yet be fully understood in terms of additional epigenetic effects, warranting further mechanistic studies. The present data support AR42 regulation of cell proliferation genes resulting in enhanced tumor cell clearance when paired with various cytotoxic agents, such as pazopanib, cisplatin, or decitabine, and the sensitization of MM cells to lenalidomide through CD44 upregulation [20,21,23,27]. Our in vitro and in vivo data presented here contribute another example of potential synergistic outcomes in a lymphoma model without any appreciable toxicity, providing support for further exploring the combination in mammalian models.

## 4. Materials and Methods

### 4.1. Treatments AR42 and FWGP

AR42 (OSU HDAC42, (S)-N-hydroxy-4-(3-methyl-2-phenylbutanamido) benzamide) was obtained from Selleckchem (S2244). Fermented wheat germ protein was prepared as described previously [4]. Briefly, wheat germ was ground to flour quality, mixed with water and baker’s yeast, and allowed to ferment at 30 °C for 48 h. The resulting slurry was centrifuged and labeled FWGE (fermented wheat germ extract). FWGP (protein) was derived from FWGE through ethanol precipitation of the proteins.

### 4.2. Cell Lines

The Burkitt B-cell lymphoma cell lines CA46 (CRL-1648), Raji (CCL-86), Ramos (CRL-1596), Daudi (CCL-213), and DG75 (CRL-2625) were purchased from the American Type Culture Collection (ATCC; Manassas, VA, USA). SuDHL-4 (ACC 495) and DOHH-2 (ACC 47) NHL cells were purchased from the Deutsche Sammlung von Mikroorganismen und Zellkulturen (DSMZ, Braunschweig, Germany). Cells were cultured in ATCC-formulated RPMI-1640 medium supplemented with 10% fetal bovine serum (FBS), 100 U/mL penicillin G, and 100 μg/mL streptomycin at 37 °C using a humidified 5% CO_2_ incubator. Media reagents were purchased from Gibco, Billings, MT, USA.

### 4.3. In Vitro Cytotoxicity Assays

Raji cells (5 × 10^4^ per sample) were plated in triplicates in 96-well plates in a volume of 100 µL. Cells were treated with FWGP (10 µg/mL or 20 µg/mL) and/or with AR42 at indicated concentrations. Untreated control cells received media only. After 24 h, 48 h, and 72 h incubation, cell viability was assessed using the CellTiter 96 AQueous One Solution Cell Proliferation Assay (Promega, G3582, Madison, WI, USA) according to the manufacturer’s instructions. Cell viability as a percentage of the untreated control was calculated as follows: [(OD490 treated − OD490 background)/(OD490 control − OD490 background) × 100]. The mean ± standard deviation (SD) of 3 separate experiments performed in triplicate is shown. Experiments were repeated with additional lymphoma cell lines CA46, Ramos, Daudi, DG75, SuDHL-4, and DOHH-2 to determine IC_50_ values after treatment with FWGP, AR42, or FWGP + AR42 after 72 h. To confirm if the combination of FWGP and AR42 yields mathematic synergy in lowering IC_50_ values, a combination index (CI) was calculated using the following formula: (IC_50_[FWGP + AR42]/IC_50_[FWGP]) + (IC_50_ [AR42 + FWGP]/IC_50_[AR42]). A CI less than 0.5 suggests a synergistic effect, while a value between 0.5 and 1.5 suggests an additive effect.

### 4.4. Flow Cytometry (FACS) for Cell Cycle Analysis

Fluorescent-activated cell sorting (FACS) was used to assess the cell cycle as previously described [37]. Raji cells, 1 × 10^6^, were treated with FWGP (20 µg/mL), AR42 (1 µM), both FWGP and AR42, or media only for 24 h. Cells were harvested and then gently re-suspended in phosphate-buffered saline (PBS). Cells were washed twice with cold PBS and then fixed overnight with ice-cold ethanol (70%, *w*/*v*). After resuspending the fixed cells with 0.25 mL of PBS, 5 µL of 10 mg/mL RNAse A (Promega, A7973) was added. Following incubation at 37 °C for 1 h, 20 µL of 1 mg/mL propidium iodide (ThermoFisher, Waltham, MA, USA) solution was added, and the cells were incubated in the dark at 4 °C for 30 min. Samples were analyzed using a Becton-Dickinson FACSCalibur (San Jose, CA, USA) cytometer reading at 488 nm. Cell cycle stages were quantified by equal gating using FlowJo software V10.09.0 as described [37].

### 4.5. Analysis of Cellular Cytotoxicity Mechanisms

FACS was used to assess the mechanisms of cancer cell killing with AR42 and FWGP. Briefly, 40k Raji cells in 96-well round-bottom plates (200 uL/well) were treated with FWGP (5 µg/mL or 20 µg/mL) with or without AR42 (0.2 µM) for 24, 48, or 72 h. Control cells were treated with PBS. Cells were washed twice with PBS and stained for 15 min with 7-AAD viability dye and PE-labeled Annexin V (eBioscience, 88-8102-74, San Diego, CA, USA), which binds phosphatidylserine on cell membranes undergoing apoptosis. Samples were acquired using a Becton-Dickinson LSR Fortessa cytometer. Single-cell lymphocytes were gated, and cell populations were defined as follows: early apoptosis (Annexin V+, 7-AAD−), late apoptosis/necrosis (Annexin V−, 7-AAD+), dead (Annexin V+, 7-AAD+), or live (Annexin V−, 7-AAD-). Percentages of cells in the combined dead and necrotic populations are presented. Gates were determined using fluorochrome isotype control beads (UltraComp eBeads, Thermo), as well as singly and doubly stained heat-killed and heat-induced apoptotic control cells.

### 4.6. RPPA Analysis and Western Blot

To determine changes in various cell survival markers, FWGP-treated Raji cell lysates were analyzed by reverse phase protein array (RPPA) using a select panel of 400+ protein targets from the MD Anderson Functional Proteomics Reverse Phase Protein Array Core facility. Briefly, Raji cells were treated with FWGP (10 µg/mL or 40 µg/mL) for 1–8 h, and cell lysates were prepared as recommended by the Proteomics core facility website. The proteins of interest were detected by probing with a specific antibody, amplifying the signal via a tyramide amplification system, and visualized via DAB colorimetric reaction. Spot intensities from the TIFF files were determined via Array-Pro Analyzer software. All processing and analysis were completed by the manufacturer and an output of expression values normalized to total protein content was provided. Protein expression values from select markers from FWGP-treated vs. untreated Raji were compared and reported.

Additionally, Raji cells were treated with 20 or 80 µg/mL of FWGP with or without 0.2 µM AR42 for 8 or 24 h to quantify activity as a deacetylase inhibitor by detection of acetylated tubulin. For in vitro immunoblots, cells were pelleted in tubes by centrifugation and washed once with cold PBS. The supernatant was aspirated and the cell pellets were lysed in 100 µL of RIPA Lysis Buffer 50 mM Tris, 150 mM NaCL, 1% Triton X-100, 0.5% Sodium Deoxycholate, 1 mM EDTA, 10 mM NaF, and 1× protease inhibitors set VII (EMD Millipore, Chicago, IL, USA). For SDS-PAGE, equal amounts of protein were loaded into precast 4–20% TGX pre-cast gradient gels (BioRad, Hercules, CA, USA) and wet transferred onto nitrocellulose membranes. Membranes were blocked in 2% Blotto (BioRad) for 1 h at room temperature and then blotted with anti-acetylated tubulin antibody (Millipore, T7451) and anti-B-actin (Sigma, A5541, St. Louis, MO, USA) overnight at 4 °C at 1:5000 in 2% Blotto. B-Actin (Sigma, A5541) primary incubation was also performed overnight at 4 °C at 1:20,000 in 2% Blotto. Primary incubation was followed by secondary incubation with HRP-conjugated antibodies (Promega, W402b), and signal detection and quantitation were performed using WesternSure Premium Chemiluminescent Substrate (Li-Cor, 926-95000, Lincoln, NL, USA) and a C-Digit scanner (Li-Cor). Signal intensity for detected acetylated tubulin was normalized to B-actin (protein loading control). Graphs were generated using GraphPad Prism 9.4.1 and Image Studio Digits 5.0 (Li-Cor).

### 4.7. In Vivo Efficacy and Toxicity in Lymphoma Xenograft Model

Female athymic nu/nu mice (6–8 weeks old) were obtained from Harlan Sprague–Dawley (Indianapolis, IN, USA) and maintained in micro-isolation cages under pathogen-free conditions in the UC Davis animal facility. Female nu/nu mice were selected for consistency with prior studies investigating FWGP. All procedures were conducted under IACUC-approved protocol according to guidelines specified by the National Institute of Health Guide for Animal Use and Care. Mice were allowed to acclimatize for at least 4 days prior to the start of any experiment. Three days before tumor cell implantation, mice received 400 rads of whole-body radiation. To establish tumors, 5 × 10^6^ Raji cells resuspended in PBS were subcutaneously implanted onto the flank of each animal. Mice were separated into the following treatment groups: PBS (400 µL, N = 7), AR42 (10 mg/kg, N = 7), FWGP (250 mg/kg, N = 7), or both AR42 and FWGP (N = 5). Mice received intraperitoneal treatment injections every other day for 45 days after tumors were established (group average = 300 mm^3^, about 2 weeks). Prior studies have demonstrated the efficacy of both agents in multiple delivery methods (orally, intravenous, intraperitoneal) [4,21,28,29]. Intraperitoneal injections were deemed appropriate due to the higher bioavailability of the agent compared to oral administration. Blood was collected from 3 mice per group on days 7, 14, 21, and 28 for blood counts and hepatotoxicity. In terms of monitoring, the UC Davis IACUC policy was followed. This mandates that mice are monitored daily and euthanized if they show signs of distress and pain (changes in coat integrity, motor activity, eating, or drinking) or disease (hind leg paralysis or bleeding), significant weight loss (>15%), or when tumors reach 1500 mm^3^ or 20 mm in any direction. No analgesics or anesthesia were used in this study. Additionally, at the end of the study, mice were euthanized by CO_2_ exposure and tumors were resected to compare volume and weight.

### 4.8. Statistical Analysis

In vitro cytotoxicity data were analyzed by a two-tailed, unpaired Student’s *t*-test. To obtain IC_50_ values, the dose–response data were fitted to a dose–response inhibition curve using the GraphPad Prism non-linear fit model. The experiments were repeated, with each experiment containing triplicate wells. FlowJo (San Diego, CA, USA) was used to analyze flow cytometry data and calculate the area under the curve corresponding to cell cycle population statistics. In vivo tumor inhibition is presented as the average tumor volume within each group over time with standard error of the mean (SEM). Statistical analysis was performed using ANOVA with the Holm–Sidak test for multiple comparisons. Significant ranges are indicated in the figure legends (* = *p* < 0.05; ** = *p* < 0.01; *** = *p* < 0.001).

### 4.9. Ethics

All animal work has been conducted according to relevant national and international guidelines under approved protocols from the University of California Davis Institutional Animal Care and Use Committee (AAALAC accreditation #000029; PHS Animal Assurance #A3433-01; USDA Registration #93-R-0433). No patient was recruited or sample collected for the sole purpose of this study.

## 5. Patents

Patent No.: 9,480,725 B2. Title: Fermented Wheat Germ Proteins (FWGP) for the Treatment of Cancer. This does not alter our adherence to PLOS ONE policies on sharing data and materials.

## Figures and Tables

**Figure 1 ijms-25-07866-f001:**
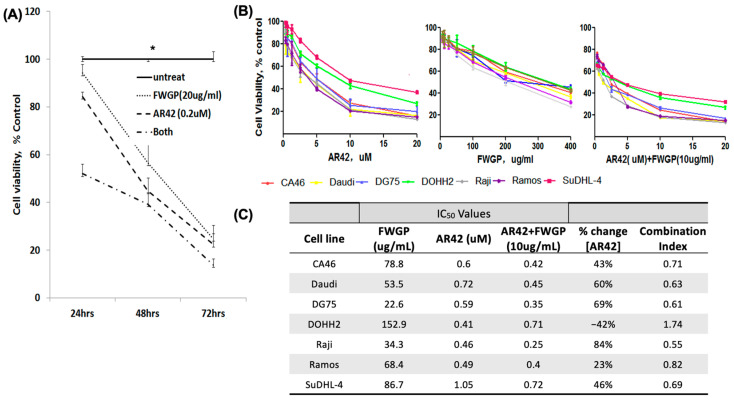
(**A**) Growth inhibition curves of Raji cells treated with FWGP (20 µg/mL), AR42 (0.2 µM), FWGP + AR42, or no treatment after 24, 48, and 72 h treatment incubation. Combination of AR42 with FWGP exhibits enhanced growth inhibition of Raji cells at all time points. Student’s *t*-test was used to determine significance * = *p* < 0.05; (**B**) Dose–response curves for AR42, FWGP, or AR42 in the presence of FWGP (10 µg/mL) were used to calculate IC_50_ values across 7 lymphoma cell lines with a non-linear fit model after log transformation of the experimental doses. Target cells (10^4^) were incubated for 72 h and cell viability was measured using an MTS assay. The mean of 3 separate experiments performed in triplicate is shown. (**C**) Combination Indices (Cis) were calculated to quantify the additive effect of combined FWGP + AR42 treatment.

**Figure 2 ijms-25-07866-f002:**
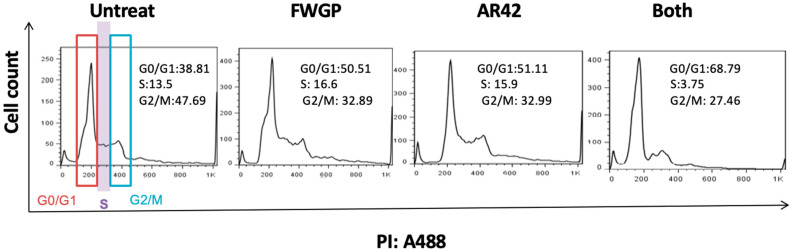
Cell cycle arrest in FWGP (20 µg/mL) and/or AR42 (1 µM)-treated Raji cells. To analyze cell cycle distribution, Raji cells were cultured with FWGP and/or AR42 for 24 h. Propidium iodide (PI)-staining was performed after 24 h of culture as described and cell cycle proportions were generated using the Watson pragmatic model in FlowJo software (V10.9.0).

**Figure 3 ijms-25-07866-f003:**
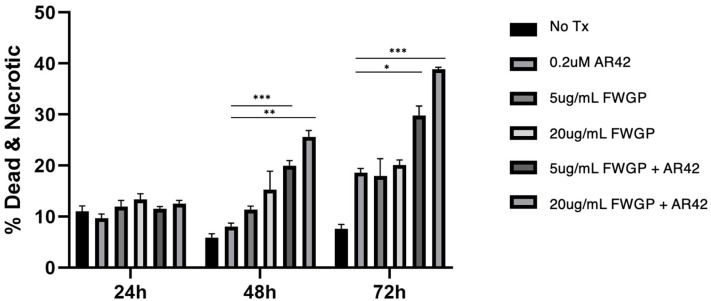
FWGP combination treatment induces a greater proportion of cell population into necrotic/dead phase. Raji cells (40 k) were treated with FWGP (5 µg/mL or 20 µg/mL) and AR42 (0.2 µM) for the time points indicated. Cells were harvested and stained with Annexin V and 7-AAD. Combined percentages of single cells in either the late apoptosis/necrotic quadrant (7-AAD+/Annexin V−) or the dead quadrant (7-AAD+/Annexin V+) are shown. Appendix A shows the acquisition gates and events captured. A 2-way ANOVA test for statistical difference was used to highlight significance, comparing the combination and AR42 monotherapy * *p* < 0.05, ** *p* < 0.005, *** *p* < 0.001.

**Figure 4 ijms-25-07866-f004:**
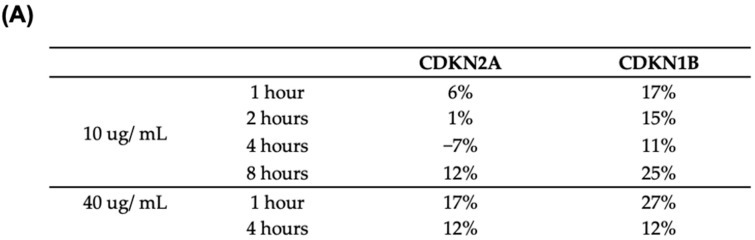
RPPA analysis of CDK inhibitor expression level and Western blot quantification histone deacetylase activity. Raji cells were incubated with FWGP for the indicated times and dosages. (**A**) Cell lysates were assayed for protein expression by Reverse Phase Protein Array. Values represent increased % expression relative to untreated control. (**B**) Acetylated tubulin was detected by Western blot in cells treated with FWGP (20 or 80 µg/mL), AR42, or FWGP + AR42 (combo). Resulting band intensities were quantified and compared; bars represent the ratio of signal intensity for detecting acetylated tubulin (HDAC activity) of treated samples to untreated control. Raw western blot images are available in Appendix A.

**Figure 5 ijms-25-07866-f005:**
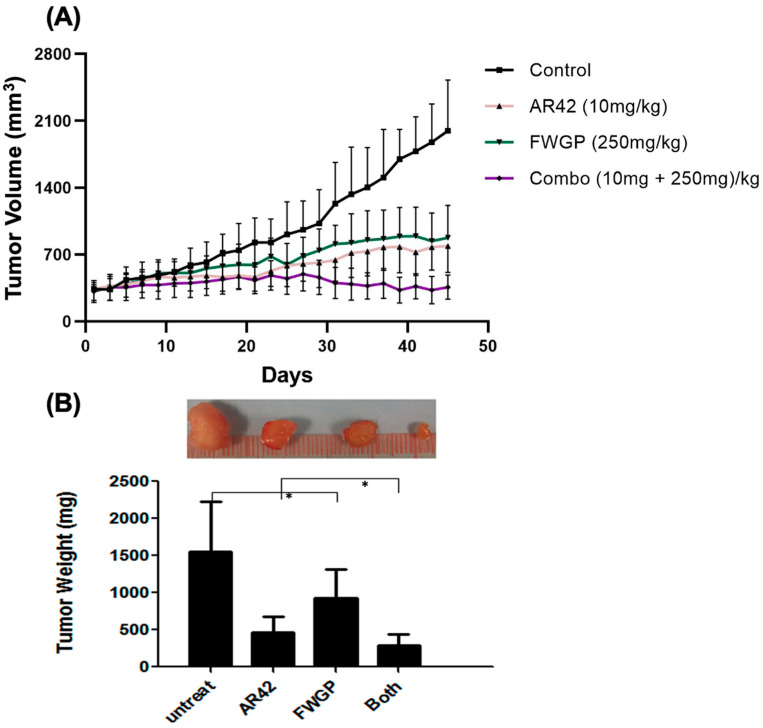
In vivo efficacy of AR42 and/or FWGP treatment in lymphoma murine xenograft models. To establish tumors, nu/nu mice were subcutaneously injected with 5 × 10^6^ Raji cells 3 days after receiving 400 rad of whole-body radiation. Tumors were allowed to grow for two weeks until they reached a group average volume of 300 mm^3^, designated as day 0. Treatment groups consisted of PBS, AR42 (10 mg/kg), and/or FWGP (250 mg/kg). All treatments were administered by intraperitoneal injection every other day for 45 days. (**A**) Tumor volume was calculated using the formula (Length × Width^2^)/2. (**B**) At the end of the study, tumors were resected from each mouse, weighed, and averaged for each group. * = *p* < 0.05.

## Data Availability

Data are contained within the article.

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
