# Peer review of "Fermented Wheat Germ Protein with Histone Deacetylase Inhibitor AR42 Demonstrates Enhanced Cytotoxicity against Lymphoma Cells In Vitro and In Vivo"

_ijms, 2024, doi:10.3390/ijms25147866_

Round 1

Reviewer 1 Report

Comments and Suggestions for Authors

In previous study, the authors discovered fermented wheat germ protein (FWGP) has similar anti-cancer effect compared to fermented wheat germ extract (FWGE).  In current study, the authors tested the efficacy of FWGP, in combination with the HDACi AR42, in lymphoma cells. The authors suggest that the combine treatment is a potentially new, minimally toxic treatment modality for lymphoma. In overall, study is too preliminary.  There are a number of papers has been published on effect of AR-42 on lymphoma as a single agent or combine with other chemotherapy drugs. The FWGP and AR42 combination treatment shows additive affect but not synergistic effect.  In addition, AR-42 is an orally active inhibitor of histone deacetylases (HDACs), it is unclear why AR42 and FWGP were administered by intraperitoneal injection in the in vivo study. Another major concern about this study is that the active protein in FWGP is not identified and injection of crude protein extract for therapy is not feasible.

Reviewer 2 Report

Comments and Suggestions for Authors

This manuscript investigated the cytotoxicity of combination treatment using fermented wheat germ protein and AR42 in lymphoma. The results show that the combination treatment has better anti-lymphoma activity than monotherapy. However, there are some issues that need to be addressed.

1.     Authors detected the cytotoxicity of FWGP, AR42, and combination treatment in Raji cells, however, whether the concentration that can lead to cancer death also have effect on normal cell is not clear. It will be better to determine the cytotoxicity of FWGP, AR42 and combination treatment in normal cells such as fibroblast or PBMC.

2.     The combination treatment in cancer therapy is widely used in lymphoma. Although authors indicated that the FWGP and AR42 combination treatment caused greater cell death than single treatment, it will be better to carry out isobologram analysis to determine whether the combination treatment has synergistic or additive effect.

3.     In Figure 4, the western blot picture was not shown. It will be better to provide the western blot pictures.

4.     In Figure 5, it will be better to plot the colony pictures and the analysis results.

Round 2

Reviewer 1 Report

Comments and Suggestions for Authors

There is no significant improvement in the revised manuscript. 
